# Effect of Additives Ag and Rare-Earth Elements Y and Sc on the Properties of Hydrogen Sensors Based on Thin SnO₂ Films during Long-Term Testing

**Nadezhda K. Maksimova [1], Aleksei V. Almaev [2,\*], Evgeniy Yu. Sevastyanov [1], Aleksandr I. Potekaev [1], Evgeniy V. Chernikov [1], Nadezhda V. Sergeychenko [1], Petr M. Korusenko [3,4] and Sergey N. Nesov [3]**

[1] Laboratory of Semiconductor Devices, Kuznetsov Siberian Physical Technical Institute, Tomsk State University, 634050 Tomsk, Russia
[2] Laboratory of Functional Electronics, Tomsk State University, 634050 Tomsk, Russia
[3] Laboratory of Nanomaterials and Heterostructures, Omsk Scientific Center of Siberian Branch, Russian Academy of Sciences, 644024 Omsk, Russia
[4] Department of Solid State Electronics, St. Petersburg State University, 198504 Saint Petersburg, Russia
[\*] Correspondence: almaev_alex@mail.ru

**Abstract:** The paper presents the results of an investigation of the nanostructure, elements, and phase composition of thin (100–140 nm) tin dioxide films obtained via magnetron sputtering and containing Ag, Y, Sc, Ag + Y, and Ag + Sc additives in the volume. Electrical and gas-sensitive characteristics of hydrogen sensors based on these films with dispersed Pt/Pd layers deposited on the surface were studied. The additives had a significant effect on the nanostructure of the films, the density of oxygen adsorption sites on the surface of tin dioxide, the band bending at the grain boundaries of tin dioxide, the resistance values in pure air, and the responses to hydrogen in the concentration range of 50–2000 ppm. During the long-term tests of most of the samples studied, there was an increase in the resistance of the sensors in clean air and in the response to hydrogen. It has been established that the joint introduction of Ag + Y into the volume of films prevents the increase in the resistance and response. For these sensors based on thin films of Pt/Pd/SnO₂:Sb, Ag, Y the responses to 100 and 1000 ppm of H₂ are 25 and 575, correspondingly, the response time at exposure to 100 and 1000 ppm of H₂ are 10 and 90 s, the recovery time at exposure to 100 and 1000 ppm of H₂ 17 and 125 s. Possible mechanisms of the effect of additives on the properties of sensors and the stability of their parameters during long-term operation were considered.

**Keywords:** hydrogen sensor; thin film; tin dioxide; silver; rare-earth elements; stability

---

## 1. Introduction

The development of hydrogen technologies requires the construction of high-speed, highly sensitive hydrogen sensors with low energy consumption and high stability of the parameters during operation. To solve this problem, it is advisable to use sensing elements based on metal oxide semiconductors whose properties can be controlled by introducing catalytic additives into the volume and depositing them on the surface. The effectiveness of such an approach was demonstrated experimentally in a number of studies conducted on films with a thicknesses of >500–1000 nm obtained by thick-film technologies [1–6]. Concomitantly, questions concerning the processes of degradation and the stability of the sensor parameters during long-term operation remain insufficiently investigated [7–9]. In this respect, there are practically no studies of thin nanocrystalline films prepared by magnetron sputtering methods that, in combination with microelectronic technology, allow a

large number of miniature sensitive elements with identical characteristics to be obtained in one technological cycle.

The noble metals Pt, Pd, and Au are most often used as catalysts. It was previously established [10,11] that the sensors based on the thin nanocrystalline antimony-doped tin dioxide films with dispersed layers of palladium and platinum deposited on the surface Pt/Pd/SnO$_2$:Sb demonstrate the highest possible response values to the effect of hydrogen from 10 to 5000 ppm and selective detection of hydrogen in mixtures of air and CO, CH$_4$, NO$_2$. In cases of films modified with gold Au/SnO$_2$:Sb, Au, the responses to H$_2$ are much lower, but selective detection of hydrogen in mixtures of air + reducing gases is more pronounced. Such devices can be used to detect hydrogen leaks in the working area of nuclear power plants [12,13] and submarines [14] at the level of 0.1 lower explosive limit (40000 ppm) and to the breath analyze for diagnosis of the lactose intolerance at the level of 20–120 ppm [15,16]. However, the peculiarity of sensors based on the Pt/Pd/SnO$_2$:Sb and Au/SnO$_2$:Sb, Au films is the instability of their parameters under prolonged exposure to hydrogen: During the tests, the resistance of the films and the response to H$_2$ increases [17]. It should be noted that under prolonged exposure to other reducing gases and vapors (CO, H$_2$S, acetone, ethanol, etc.), the sensor sensitivity decreases [18–20].

The effect of additives of 3D-transition metals Co and Ni introduced into the volume of the gold-modified SnO$_2$ films on the electro-physical and gas-sensitive properties, as well as on the stability of parameters of hydrogen sensors, was investigated in [21]. It was shown that during testing, a slight increase in the resistance in pure air and response occurs. Complete stabilization of the film response, in particular at high concentrations (>100 ppm) of hydrogen, cannot be achieved.

It is necessary to study the effect of rare-earth element additives on the characteristics of gas sensors. The published information on the properties of such structures is limited and usually refers to samples obtained using thick-film technology [22–26]. No data for thin films with additives of rare-earth elements fabricated by the magnetron sputtering methods are known to the authors. Our preliminary studies have shown that to stabilize the characteristics of thin-film hydrogen sensors, it is advisable to use tin dioxide films with the joint addition of yttrium and silver into the volume. A sufficient number of studies have been devoted to the catalytic properties of silver [27–31], but the questions regarding Ag's influence on the properties of gas sensors remain controversial.

The purpose of this work is to investigate the nanostructure, element and phase composition, electrical and gas-sensitive characteristics, and parameters stability of H$_2$ sensors based on thin nanocrystalline SnO$_2$ films with dispersed Pt/Pd layers deposited on the surface and Ag, Y, Sc, Ag + Y, and Ag + Sc additives in the volume.

## 2. Materials and Methods

The SnO$_2$ thin films were produced in a magnetron A-500 Edwards using direct-current (DC) sputtering in an oxygen-argon plasma of the target, which is a tin-antimony alloy (0.51 at % of Sb). The diameter of the target was 3 inches and the thickness was $\frac{1}{4}$ inches. Commercially available 150 μm thick polished wafers of the polycrystalline sapphire (Al$_2$O$_3$) were used as substrates. The diameter of the sapphire substrates was 30 mm. The conditions of sputtering of all films were: The working pressure in the chamber $8.5 \times 10^{-3}$ mbar and power 70 W, the substrate was at room temperature. The oxygen concentration in the mixture Ar + O$_2$ was maintained at 56.1 ± 0.5 vol.%. The distance between the target and the substrate was 70 mm. The deposition time of SnO$_2$ was 24 min to form needing thickness of the films.

The results obtained earlier showed that the sensors based on thin films of pure SnO$_2$ produced by DC magnetron sputtering of the tin target are characterized by a resistance of about $10^9$–$10^{12}$ Ω. Measurements of the characteristics of such high-resistance structures are strongly influenced by noise. Therefore, a target that is an alloy of tin and antimony was used to create gas sensors. The antimony impurity creates donor centers of electrons in tin dioxide and helps reduce the resistance of thin films to 1–10 MΩ at operating temperatures of gas sensors [11].

To introduce additives Ag, Y, and Sc into the bulk of the films, pieces of the corresponding metals were placed on the surface of the target sprayed part (Figure 1). The ultrathin catalytic Pt/Pd layers were deposited on the surface of the tin dioxide with the additives of metals in the bulk by DC-magnetron sputtering of metallic target (Pt and Pd). The conditions of Pt/Pd layers deposition: The working pressure in the chamber $8.5 \times 10^{-3}$ mbar and power 70 W, the wafers were at room temperature. The palladium was first deposited for 15 s, then the platinum for the same time. Such the deposition time of catalytic additives on the surface of tin dioxide allows to form the ultradisperse layers of platinum and palladium that maximize the sensor responses to hydrogen [11].

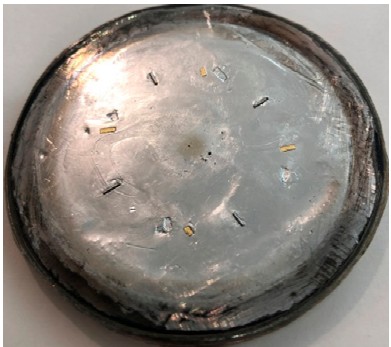

**Figure 1.** Photo of the tin target with pieces of the metals introducing into the bulk of $SnO_2$ film. The tin target with the addition of gold and nickel is shown for the best contrast.

The heater on the back side of the substrate and electrical contacts to the sensitive layers (Figure 2) were formed by spraying platinum on the sapphire wafers heated to 773 K with subsequent photolithography engraving before magnetron deposition of the $SnO_2$ films. Two photolithography operations were used to form the sensitive elements of a defined shape and size.

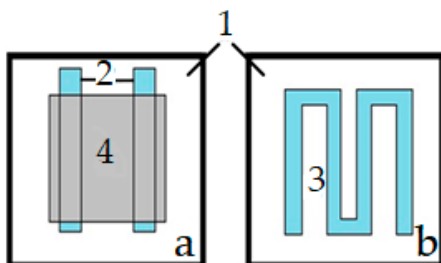

**Figure 2.** The gas sensor on the side of the sensitive layer (**a**) and the heater (**b**): 1-sapphire substrate; 2-platinum electrodes; 3-heater; 4-$SnO_2$ film.

All fabricated wafers with the $SnO_2$ films were subjected to stabilizing annealing in the atmosphere at 723 K for 24 h. After cutting, sensors 0.7 mm × 0.7 mm in size (with the area of the sensitive layer of 0.3 mm × 0.3 mm) were assembled into TO-8 cans (Figure 3). Up to 500 sensors were obtained on one substrate with a diameter of 30 mm.

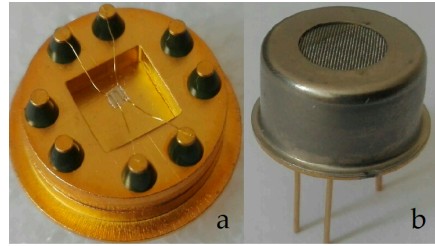

**Figure 3.** (**a**) Photo of the sensor assembled into the TO-8 can and (**b**) sensor with a protective cap.

We introduced the following numbering of series and notations for films with various additives in the volume:

- Pt/Pd/SnO$_2$:Sb;
- Pt/Pd/SnO$_2$:Sb, Ag;
- Pt/Pd/SnO$_2$:Sb, Y;
- Pt/Pd/SnO$_2$:Sb, Sc;
- Pt/Pd/SnO$_2$:Sb, Ag, Y;
- Pt/Pd/SnO$_2$:Sb, Ag, Sc.

To study the thickness, microrelief, and grain sizes of the thin films, the atomic-force microscopy (AFM, Solver HV, NT-MDT, Moscow, Russia) method was used. The refinement of the element composition of the films was carried out using a scanning Auger microprobe equipped with an Ar$^+$-ion sputtering system (AES) and a X-ray photoelectron spectroscopy (XPS) on the specially manufactured films of SnO$_2$:Sb, Ag; SnO$_2$:Sb, Ag, Y; SnO$_2$:Sb, Sc; and SnO$_2$:Sb, Ag, Sc.

The AES method was realized on a Shkhuna-2 installation (Electron, Ryazan, Russia) containing a heated ultrahigh vacuum chamber in which an Auger analyzer (of a cylindrical mirror analyzer type) with an energy resolution of 0.1% was placed. An electron source forming a probing beam with a diameter of 1 μm was embedded coaxially into the analyzer. The electron energy was 3 keV. An argon ion beam with a diameter of 1.5 mm was used to spray the surface of the sample under study. The energy spectra of the Auger electrons were recorded in scanning mode by an electron beam, and the scanning area was 100 μm × 100 μm. This method makes it possible to detect all of the elements (except hydrogen and helium) whose concentrations are higher than 1 at %.

The XPS method was implemented on the Surface Science Center (Riber, Bezons, France) analytical complex. Al Kα radiation ($hv$ = 1487 eV) was used to excite X-ray spectra. The XPS spectra were obtained in an ultrahigh vacuum (~10$^{-9}$ Torr) using a two-stage cylindrical mirror analyzer EA 150 (Riber, Bezons, France). The diameter of the X-ray beam was ~5 mm, the power of source was 240 W. The energy resolution for the spectra was ~0.1 eV. A layered XPS analysis was performed directly in the spectrometer chamber. The coating layer was etched by a beam of argon ions with an average energy of 3 keV at a pressure of ~10$^{-5}$ Torr in the spectrometer chamber. The etching rate was ~1–3 nm/min.

Measurement of the absorption spectra was conducted using a CM2203 spectrophotometer (Solar, Minsk, Belarus). To determine the phase state of silver (Ag, AgO, or Ag$_2$O) in the tin dioxide, the UV-vis spectroscopy method was used. Based on the analysis of the optical absorption edges, the band gaps in the films with the Ag and Ag + Y additions were estimated.

The conductance $G_0$ (resistance $R_0$) of the films in pure air and similar $G_1$ ($R_1$) parameters upon exposure to H$_2$ were measured as functions of the operating temperature T and the hydrogen concentration $n$ in air using a specially designed stand [11,17–21] (Figure 4). The ratio $G_1/G_0$ was obtained for the adsorption response. The time in which conductivity reached 0.9 of the stationary $G_{st}$ value was considered the response time $t_{res}$. The time in which conductivity reached 1.1 of the stationary $G_{0st}$ value after gas exposure was considered the recovery time $t_{rec}$.

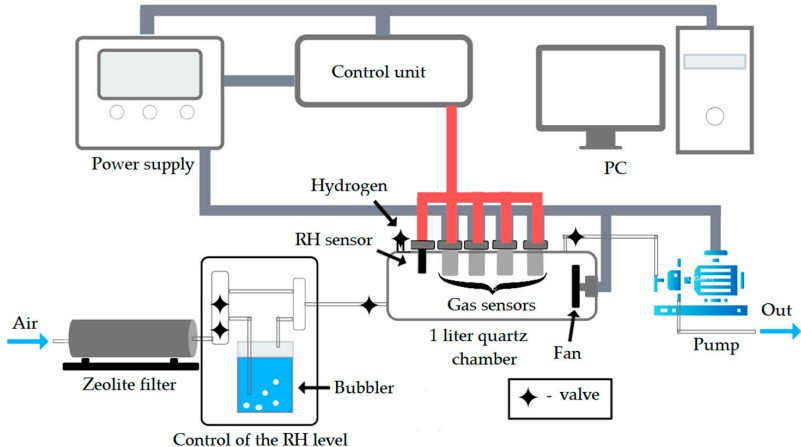

**Figure 4.** The schematic block diagram of the measuring stand.

To measure the characteristics, four sensors were simultaneously placed in a 1 liter quartz chamber equipped with a fan. Two air flows with controlled rates were pumped through the chamber to control the humidity level: One flow was dried with zeolite and the second flow was damped by bubbling. Then the chamber was encapsulated. Humidity control was maintained using a capacitive sensor HIH 4000 by Honeywell (Charlotte, NC, USA) located in the chamber. The necessary gas concentration was input with a syringe dispenser, which created the necessary composition of the gas–air mixture. As a source of hydrogen, a balloon of a mixture of hydrogen (4.02 vol.%) and air was used. The hydrogen concentration was increased by addition of the next portion of gas to measure the concentration dependencies of the sensor response. The applied voltage to the $SnO_2$ thin films was 2.5 V. After the measurement, the chamber was pumped with pure air at a required humidity level. This paper presents experimental data obtained at an average relative humidity level of $RH = 30\%–35\%$ at room temperature in the chamber.

In addition, experiments with the stand including the gas flow meters Bronkhorst (Ruurlo, The Netherlands) were carried out to control the content of the hydrogen in using a mixture of the nitrogen and the oxygen of high purity as air. The total gas flow rate through the measuring chamber with volume 0.6 L was 3.0 L/min. The values of sensor conductivity in using two types of stand in the same conditions were the similar. However, the second system is far from the real conditions of sensors usage.

To study the stability of the sensor parameters in the course of prolonged operation under the action of hydrogen, the concentration dependences of the response were measured every 2–4 days and more (up to 30–60) days in the concentration range 50–1000 ppm of $H_2$. The time dependencies of the sensor's conductance at exposure to 1000 ppm of $H_2$ after the long-term tests are shown in Figure 5. Changes in the conductivity at exposure to gas are reversible.

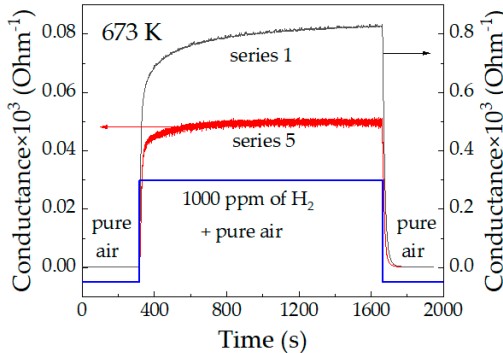

**Figure 5.** The time dependencies of the sensor's conductance at exposure to 1000 ppm.

## 3. Results

### 3.1. Nanostructure and Composition of the Films

Figure 6a shows the electron Auger spectra from different depths of the SnO$_2$:Sb, Ag, Y films and Figure 6b,c demonstrates the profiles of relative concentrations of the detectable elements. For clarity, in Figure 6a, the energy regions of 250–325 and 425–450 eV, to which the signals from the elements with low concentrations (carbon, silver, antimony) correspond, are separated by vertical lines, within which the amplitudes increase fivefold. On the surface and to a depth of 5 nm, carbon is observed, which is usually adsorbed from the atmosphere. In the energy region corresponding to yttrium, no signal is detected, which indicates a low (<1 at %) concentration of Y in the tin dioxide film. In the interval of energy from 320 to 420 eV, superposition of the Ag and Sn spectra occurs. Relative concentrations of Sn and O correspond to SnO$_2$ (Figure 6b), the Ag content is about 1.2 at %, and the concentration of antimony (0.9–1 at %) exceeds the content of this impurity in the target (0.51 at %) (Figure 6c). The Sc content is about 3.8–4 at %. It should be noted that the AES method is semiquantitative.

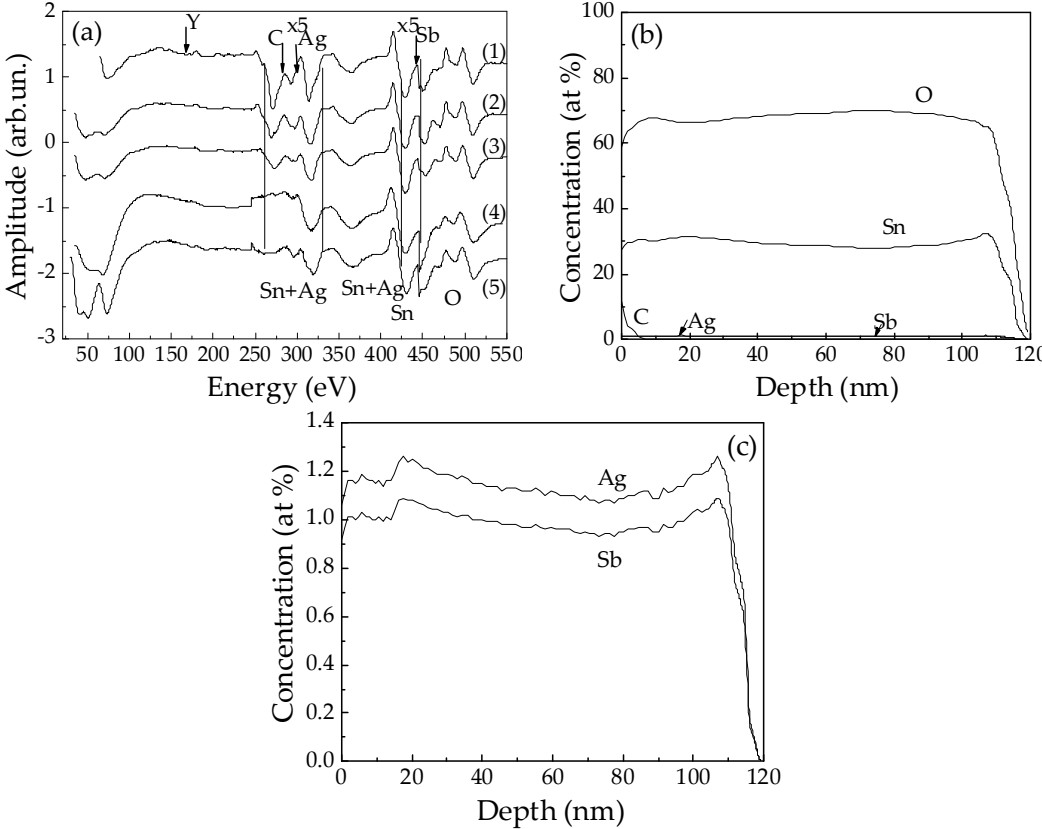

**Figure 6.** (**a**) Auger spectra of the SnO$_2$:Sb, Ag, Y film with respect to electron kinetic energy in the near-surface region (1) and for the different layer depth, nm: 3 (2), 5 (3), 30 (4), and 90 (5). (**b**) In-depth profiles of all detected elements concentration ratio in SnO$_2$:Sb, Ag, Y layer bulk and in-depth profiles of Ag and Sb concentration (**c**).

The XPS spectrum is dominated by the lines of Sn and O. There are also low intensity lines Ag, Sc, Y. The most intensive lines of the corresponding elements were chosen for quantitative analysis. The atomic sensitivity factor was determined in accordance with [32]. To average XPS data, spectra were measured at three different points on the surface. The analysis of the samples composition was carried out after etching for 40–60 min, i.e., the data corresponded to the volume of films. The values of additive concentrations obtained from XPS analysis 1.4–1.5 at % for Ag, 0.3–0.34 at % for Y, and 3.0–3.4 at % for Sc are in good agreement with the AES data.

Chemical analysis of the elements was carried out on the spectra of the most intense lines measured with high energy resolution (Figure 7). Table 1 presents the values of binding energy (BE) of the core levels Sn $3d_{5/2}$, O $1s$, Ag $3d_{5/2}$, Sc $2p_{3/2}$, and Y $3d_{5/2}$. According to the literature data [32,33] in tin dioxide the positions of maxima of Sn $3d$ and O $1s$ lines lie in the ranges 486.2–487.2 and 530–531 eV, respectively. For the films studied in this paper, the binding energies (Table 1) correspond to these fundamental values. It is important to note that in the case of films containing silver, the values of BE Sn $3d_{5/2}$ and O $1s$ coincide with each other and are lower by ~0.3 eV and ~0.8–0.9 eV than in the sample SnO$_2$:Sb, Sc without the addition of silver (Table 1). These results are consistent with data [31] according to which the binding energy of Sn $3d$ and O $1s$ in SnO$_2$:Ag films decreases by 0.5–0.7 eV compared to pure SnO$_2$.

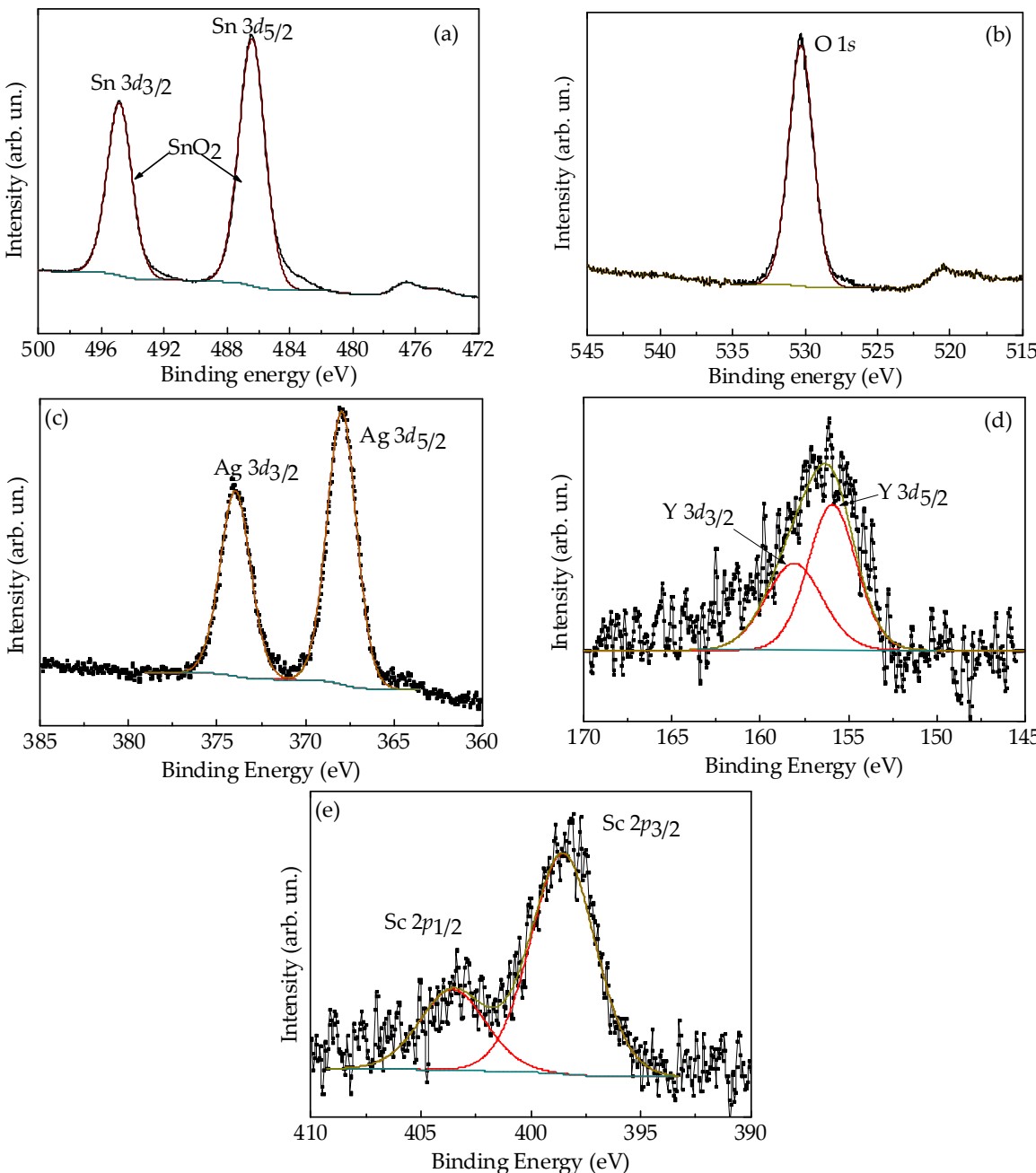

**Figure 7.** XPS spectra for (**a**) Sn $3d$, (**b**) O $1s$, (**c**) Ag $3d$, (**d**) Y $3d$, and (**e**) Sc $2p$.

**Table 1.** Spectral characteristics of the levels of Sn, O, Ag, Sc, and Y.

| The Core Levels | Composition of the Films | | |
| --- | --- | --- | --- |
| | $SnO_2$:Sb, Sc | $SnO_2$:Sb, Ag, Y | $SnO_2$:Sb, Ag, Sc |
| | Bending Energy (eV) | | |
| Sn $3d_{5/2}$ | 486.7 | 486.4 | 486.4 |
| O $1s$ | 531.0 | 530.2 | 529.8 |
| Ag $3d_{5/2}$ | – | 368.1 | 368.0 |
| Sc $2p_{3/2}$ | 398.5 | – | 398.5 |
| Y $3d_{5/2}$ | – | 155.8 | – |

Figure 7c depicts the XPS spectra taken from the Ag $3d$ region. The position of the maxima of Ag $3d$ lines is 368 eV ($3d_{3/2}$) and 374 eV ($3d_{5/2}$) as well as spin-orbital splitting correspond to the silver in the metallic state ($Ag^0$) [34]. The core levels Y $2p$ and Sc $2p$ spectra can be Gaussian fitted into two components. The binding energy of Y $3d_{5/2}$ (155.8 eV) and Y $3d_{3/2}$ (158.0 eV) correspond to $Y^0$ (Table 1) [26]. The binding energy peaks located at 398.5 and 403.4 eV in Figure 7e correspond to Sc $2p_{3/2}$ and Sc $2p_{1/2}$ lines for metallic scandium [35]. The layers of series 1, 5, and 6 with a thickness of 105–110 nm contained nanocrystallites with dimensions $d_1 \approx 8$–30 nm and $d_2$ from 120 to 200 nm (Table 2). The nanostructure of the films without additives differed slightly from that of the films containing two additives of Ag + Y and Ag + Sc. According to the AFM data, the thicknesses of the films of series 2 and 3 were 110–130 nm. For the films of series 2 and 4 with the addition of only Ag or Sc, the presence of nanocrystals with dimensions of 30–60 nm was characteristic. In the films with the addition of Y (series 3), larger crystallites of 200–350 nm in size were observed (Table 2).

**Table 2.** The main parameters of the freshly prepared sensors of the different series.

| Series | 1 Pt/Pd/$SnO_2$:Sb | 2 Pt/Pd/$SnO_2$:Sb, Ag | 3 Pt/Pd/$SnO_2$:Sb, Y |
| --- | --- | --- | --- |
| $d_1$ (nm) | 18–20 | 30–60 | 200–350 |
| $d_2$ (nm) | 70–100 | – | – |
| $R_0$ (MΩ) | 1.5–5 | 2.5–3.7 | 5–15 |
| $T_{max}$ (K) | 670–690 | 570–600 | 690–730 |
| $R_0^*$ (MΩ) | 0.5–0.7 | 1.5–2.5 | 3.5–4.5 |
| $G_1/G_0^*$ | 16–18 | 19–30 | 18–20 |
| $G_1/G_0^{**}$ | 170–180 | 290–330 | 42–45 |
| $e\varphi_s$(eV) | 0.40 | 0.44 | 0.46 |
| **Series** | **4 Pt/Pd/$SnO_2$:Sb, Sc** | **5 Pt/Pd/$SnO_2$:Sb, Ag, Y** | **6 Pt/Pd/$SnO_2$:Sb, Ag, Sc** |
| $d_1$ (nm) | 40–60 | 18–30 | 20–40 |
| $d_2$ (nm) | – | 120–200 | 120–210 |
| $R_0$ (MΩ) | 7–8 | 26–30 | – |
| $T_{max}$ (K) | 620–630 | 670–690 | 620–680 |
| $R_0^*$ (MΩ) | 0.17 | 5.9–6.5 | 15–17 |
| $G_1/G_0^*$ | 2.4–3.5 | 30–35 | 10–13 |
| $G_1/G_0^{**}$ | 10–17 | 580–590 | 230–240 |
| $e\varphi_s$(eV) | 0.52 | 0.64 | 0.74 |

The values of $R_0$ were measured at a temperature of 300 K, $R_0^*$ and the responses $G_1/G_0$ were measured at an operating temperature of 670 K at hydrogen concentrations of 100 ppm ($G_1/G_0^*$) and 1000 ppm ($G_1/G_0^{**}$).

Figure 8 shows the absorption spectra in the spectral range of 300–700 nm in 1 nm steps obtained from plates placed in a holder for solid samples. A sapphire substrate was used as the zero line. The spectra of the non-heat treated structures (Figure 8) were characteristic for amorphous films. In the sample of series 2 with the addition of only Ag, the surface plasmon resonance (SPR) band was weakly expressed and had a maximum at 375 nm. Silver was apparently present in the ionic form of $Ag^+$ corresponding to $Ag_2O$ oxide.

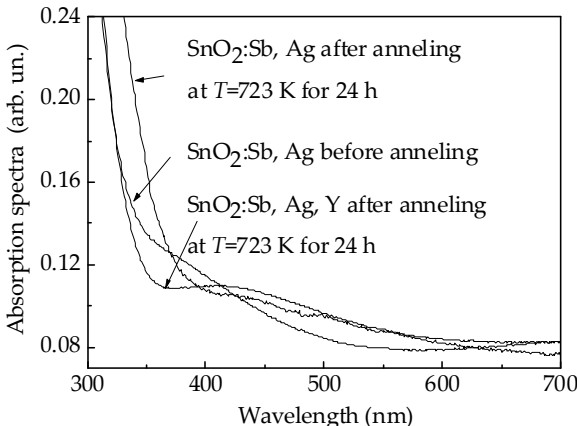

**Figure 8.** Optical absorption spectra of the thin tin dioxide films $SnO_2$:Sb, Ag before annealing and $SnO_2$:Sb, Ag as well as $SnO_2$:Sb, Ag, Y after annealing at $T = 723$ K for 24 h.

In the process of thermal annealing, polycrystalline tin dioxide films with the Ag and Ag + Y additives formed in the absorption spectra of which the SPR of silver distinctly manifested itself in the maximum at 412 nm (Figure 8). The SPR bands were broadened and "blurred," indicating the presence of polydisperse metallic $Ag^0$ nanoparticles with the sizes of up to 10 nm or more that apparently segregated on the surface of the tin dioxide nanocrystals. It should be noted that the authors of [29] reported on the metallic state of silver on the basis of the observation of the SPR band in the absorption spectra of thin (30–40 nm) films obtained via magnetron sputtering in an argon plasma of a Sn target with the Ag pieces placed on the surface. The films were subjected to oxidation in $O_2$ at 450–800 °C for 10–30 min.

The absorption edges for the films of series 2 and 5 subjected to the thermal annealing (Figure 9) contained two linear regions, the extrapolation of which to zero absorption suggested the existence of two values of the forbidden band width in the investigated polycrystalline films: $E_g = 3.66$ and 3.94 eV for the samples with the addition of Ag, and $E_g = 3.76$ and 3.88 eV for the samples with the addition of Ag + Y. Single-crystal tin dioxide is characterized by $E_g = 3.6$ eV [36]. In nanoscale thin $SnO_2$ films, variation of the band gap in the range of 3.6–4.3 eV is possible depending on the film thickness and dimensions of nanocrystallites [37–40]. It is obvious that the values of $E_g$ obtained for the Pt/Pd/$SnO_2$:Sb, Ag and Pt/Pd/$SnO_2$:Sb, Ag, Y films were within the typical limits for nanocrystalline $SnO_2$ samples.

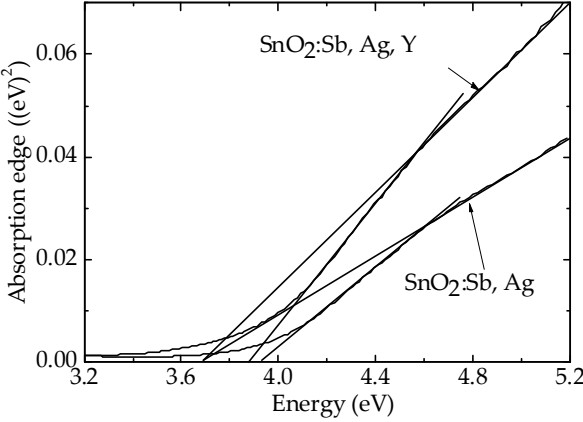

**Figure 9.** Optical absorption edges of the thin tin dioxide films $SnO_2$:Sb, Ag and $SnO_2$:Sb, Ag, Y after annealing at $T = 723$ K for 24 h.

### 3.2. Characterization of the Sensors

In Table 2 and Figure 10 the electrical and gas-sensitive properties of the freshly prepared films with various additives that were not subjected to long-term tests under the action of hydrogen are presented. Compared to the data for series 1, the values of the resistance $R_0$ of the samples of series 2 measured at 300 K in pure air did not change substantially after the addition of Ag. For the sensors with additives of only Y or Sc, $R_0$ increased slightly. The joint introduction of silver and yttrium (series 5) or scandium (series 6) contributed to a significant increase in the resistance in pure air. The resistance $R_0$ of the sensors of series 5 increased by almost an order of magnitude (up to 26–30 MΩ) (Table 2). For the sensors with the Sc + Ag additives, the most significant increase in $R_0$ was observed. In this case, noises appeared that made it difficult to measure this parameter in the region of low temperatures (300–400 K). An increase in the resistance of the sensors in pure air and the high responses to hydrogen could indicate the increase in the density of the chemisorbed oxygen on the surface of the tin dioxide and the width of the space charge region (SCR).

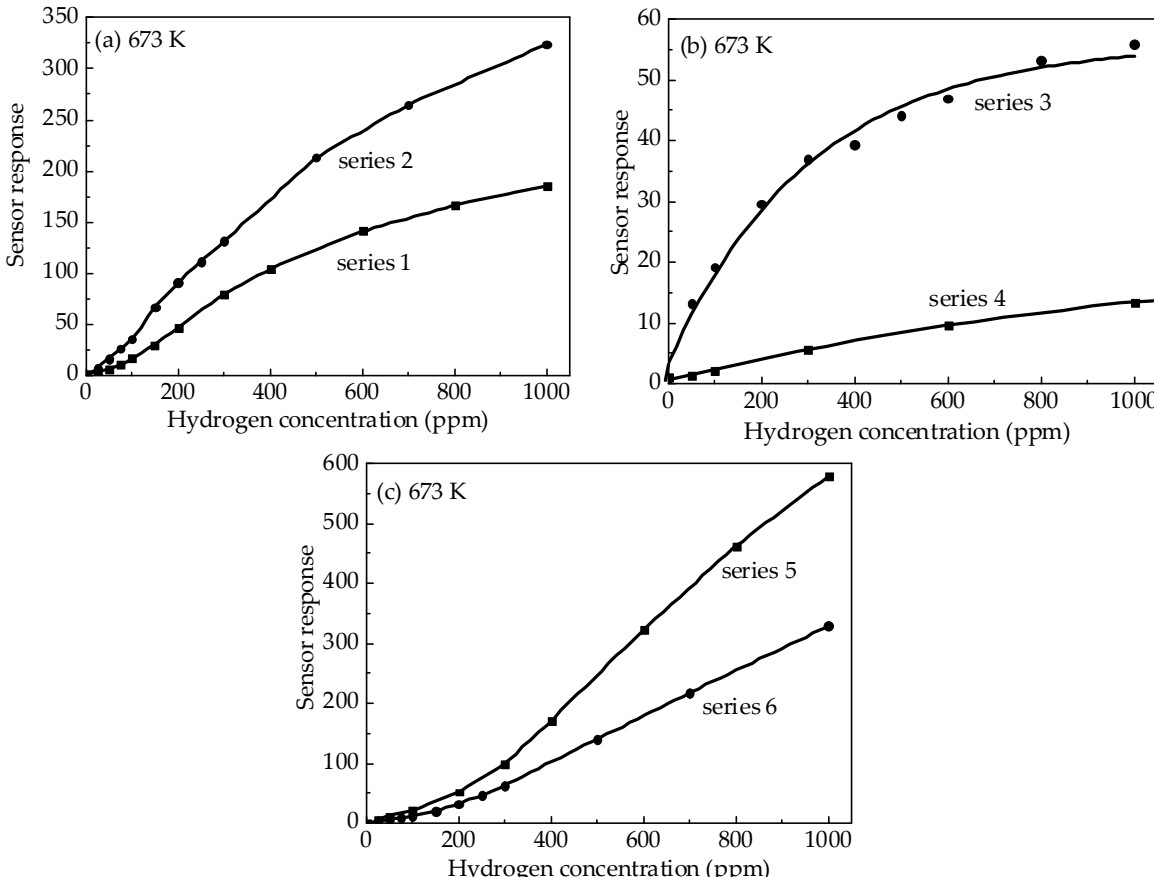

**Figure 10.** (**a**–**c**) Concentration dependencies of the response of the sensors with various additives.

The type of additives in the volume and on the surface of the thin films of the tin dioxide had a significant effect on the gas-sensitive properties of the sensors. The dependences of the adsorption response on the operating temperature for all of the samples studied were curves with a maximum at a temperature $T_{max}$. The values of $T_{max}$ were on average 670–730 K for the sensors of series 1, 3, and 5 (Table 2). The addition of Ag (series 2) reduced $T_{max}$ to 570–600 K, which was consistent with the literature data [27]. The intermediate values of $T_{max}$ = 620–680 K were characteristic for the samples with the additions of Sc (series 4) and Ag + Sc (series 6). Further measurements of $R_0$ and $G_1/G_0$ were performed at 670 K since this operating temperature provides sufficiently high response values and a low response time for all types of sensors: $t_{res}$ at exposure to 100 ppm of $H_2$ did not exceed 10–15 s,

and $t_{res}$ at exposure to 100 ppm of $H_2$ did not exceed 190 s. It is worth noting that the recovery time after exposure to 100 ppm of $H_2$ for all types of sensors did not exceed 30 s, and 125 s after exposure to 1000 ppm of $H_2$. The responses of the studied samples are higher than the responses of the hydrogen sensors based on other materials [13,41]. Therefore, the sensor response equal to 1.1 was chosen as the detection limit. The experimentally determined value of the detection limit for freshly prepared sensors of series 1–3 and 6 was ~1 ppm, 15 ppm for sensors of series 4, and 0.5 ppm for sensors of series 5.

Figure 10 compares the dependences of the responses of the different sensors on the concentration of hydrogen. Table 2 shows the values of $G_1/G_0$ at fixed concentrations of 100 and 1000 ppm of $H_2$. The concentration dependences of the response of sensors without additives in the volume (series 1) with the addition of Ag (series 2) as well as with two additives Ag + Y (series 5) and Ag + Sc (series 6) were characterized by the presence of an exponential growth at low $H_2$ concentrations (Figure 10a,c) and high values of the response to $H_2$. The introduction of only rare-earth elements Y (series 3) or Sc (series 4) into the volume of the films changed the form of the dependences of $G_1/G_0$ on *n*: They became sublinear (Figure 10b). The response values of these samples were much lower (especially for the samples with the addition of scandium).

The results of studying the effect of additives in the volume of the tin dioxide on the change in the characteristics of the sensors during long-term tests under periodic exposure to hydrogen are interesting. The main parameters of the samples after the tests are provided in Table 3. The dependences of the response to 1000 ppm $H_2$ on the duration of operation are shown in Figure 11. In the films of series 2 with the addition of silver in the volume, a sharper growth of the resistance measured in the pure air and the response to both low (100 ppm) and high (1000 ppm) hydrogen concentrations was observed compared to the samples of the other series.

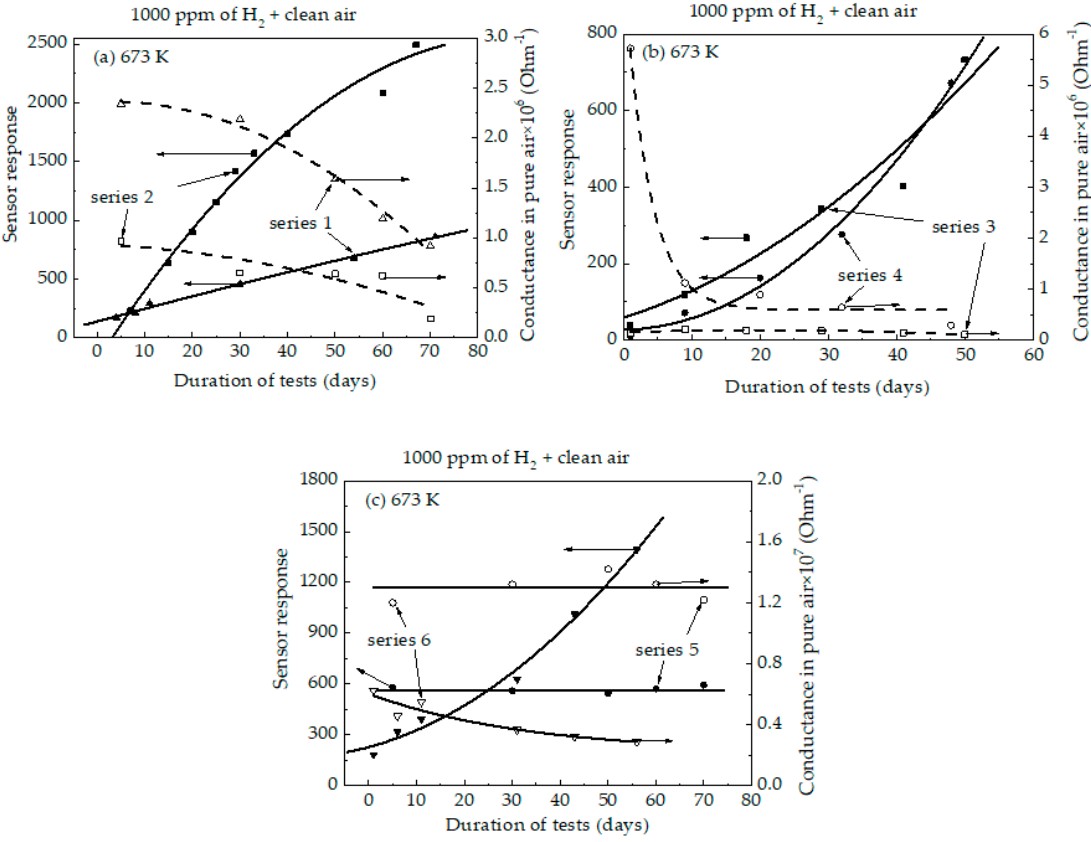

**Figure 11.** (a–c) The effect of the duration of tests on the response to 1000 ppm of $H_2$.

**Table 3.** The main parameters of the sensors of the different series after the completion of the tests.

| Series | 1 Pt/Pd/SnO$_2$:Sb | 2Pt/Pd/SnO$_2$:Sb, Ag | 3 Pt/Pd/SnO$_2$:Sb, Y |
|---|---|---|---|
| $t$ (days) | 600 | 67 | 50 |
| $T_{max}$ (K) | 670–690 | 640–650 | 690–730 |
| $R_0$* (MΩ) | 1.5–3.5 | 9.1–10 | 3.5–4.5 |
| $G_1/G_0$* | 14–16 | 130–139 | 18–20 |
| $G_1/G_0$** | 840–860 | 2480–2490 | 621–730 |
| $e\varphi_s$(eV) | 0.56 | 0.71 | 0.71 |
| **Series** | **4 Pt/Pd/SnO$_2$:Sb, Sc** | **5 Pt/Pd/SnO$_2$:Sb, Ag, Y** | **6 Pt/Pd/SnO$_2$:Sb, Ag, Sc** |
| $t$ (days) | 590 | 70 | 50 |
| $T_{max}$ (K) | 620–680 | 670–690 | 670–690 |
| $R_0$* (MΩ) | 1.2 | 6.2–7.1 | 33–35 |
| $G_1/G_0$* | 5.5–7.6 | 30–35 | 46–54 |
| $G_1/G_0$** | 640–711 | 590–597 | 1500–1600 |
| $e\varphi_s$(eV) | 0.66 | 0.72 | 0.88 |

The values of $R_0$* and the responses $G_1/G_0$ were measured at an operating temperature of 670 K at hydrogen concentrations of 100 ppm ($G_1/G_0$*) and 1000 ppm ($G_1/G_0$**).

The temperature $T_{max}$ shifts to higher temperatures of 640–650 K. Similar but less pronounced changes occurred in the samples with the Sc (series 4) and Ag + Sc (series 6) additives. For the films without additives (series 1) as well as those under the introduction of yttrium (series 3) or Ag + Y (series 5) into the volume, the responses to low (50–100 ppm) hydrogen concentrations practically did not change during all of the tests. The values of $R_0$ for the films without additives (series 1) changed slightly during all of the tests. Only the responses of the sensors with the Ag + Y additives (series 5) were stable at all studied concentrations of H$_2$ (Table 3 and Figure 11).

## 4. Discussion

### 4.1. Main Physical Models of Sensors

In real polycrystalline samples, $G_0 = G_{0b} + G_{0ch}$, where $G_{0b}$, $G_{0ch}$ are the over-barrier and the channel components of the conductivity. In the case of $G_{0b} >> G_{0ch}$, to travel from one nanocrystal to another, an electron must overcome an energy barrier, the value of which is $e\varphi_s \sim N_i^2$, where $N_i$ is the surface density of chemisorbed oxygen ions. The expression for $G_{0b}$ is:

$$G_{0b}(T) = G_{00}(T) \cdot \exp[-e\varphi_s(T)/kT] \tag{1}$$

where $G_{00}$ is the value that under the action of hydrogen changes slightly [42,43]. $G_{1b}/G_{0b}$ is proportional to the exp ($e\varphi_s$). In the region of low concentrations H$_2$ (<300 ppm) an exponential dependence of the response on $n$ was observed. An analysis of data in Figure 10a,c shows that for the films doped only with antimony impurity, as well as for those additionally containing the additives of Ag, Ag + Y, and Ag + Sc, the over-barrier conductivity played a decisive role.

In the case of the predominant role of the channel component of the conductivity ($G_{0b} << G_{0ch}$) the value of $G_{0ch}$ was proportional to (1–2$d_0/d_B$) [43], where $d_0$ is the width of the space-charge region and $d_B$ is the thickness of the conductivity bridges. The effect of hydrogen led to a decrease in the ratio of 2$d_0/d_B$ and $d_0 \sim N_i$, the dependence of $G_{1ch}/G_{0ch}$ on $n$ was sublinear, and the increase of the response with increasing hydrogen concentration was much weaker than for $G_{1b}/G_{0b}$. This model satisfactorily describes the characteristics of the sensors with the additives of only rare-earth elements Y and Sc (Table 2 and Figure 10b).

### 4.2. Method for Determining the Energy Band Bending

Papers [10,42–45] reported that to determine the energy band bending, it is necessary to know the stationary values of $G_0$ at two temperatures $T_1$ and $T_2$, but for one value of $e\varphi_s$; refer to (1). This can be

realized in the thermo-cyclic mode [10]. The formula for determining $e\varphi_s$ follows from the analysis of the ratio of conductivities $G_0$ ($T_2$) and $G_0$ ($T_1$) as well as from the temperature dependence of $G_{00}$. Thus, for $e\varphi_s$:

$$e\varphi_s = [kT_1T_2/(T_2-T_1)]\cdot\ln[(G_0(T_2)/G_0(T_1))\cdot(T_2/T_1)^{0.75}] + kT_2. \qquad (2)$$

The following temperatures and durations of the cooling and heating cycles were used for all of the sensors studied: $T_1 = 473$ K ($t_1 = 6$ s) and $T_2 = 673$ K ($t_2 = 8$ s). The justification of this choice of temperatures and durations and a detailed analysis of the conductivity-time profiles (CTP) $G_0(t)$ are presented in [10]. It should be emphasized that the details of the CTP of the sensors were determined by the type of additives and could change after long-term tests (Figure 12).

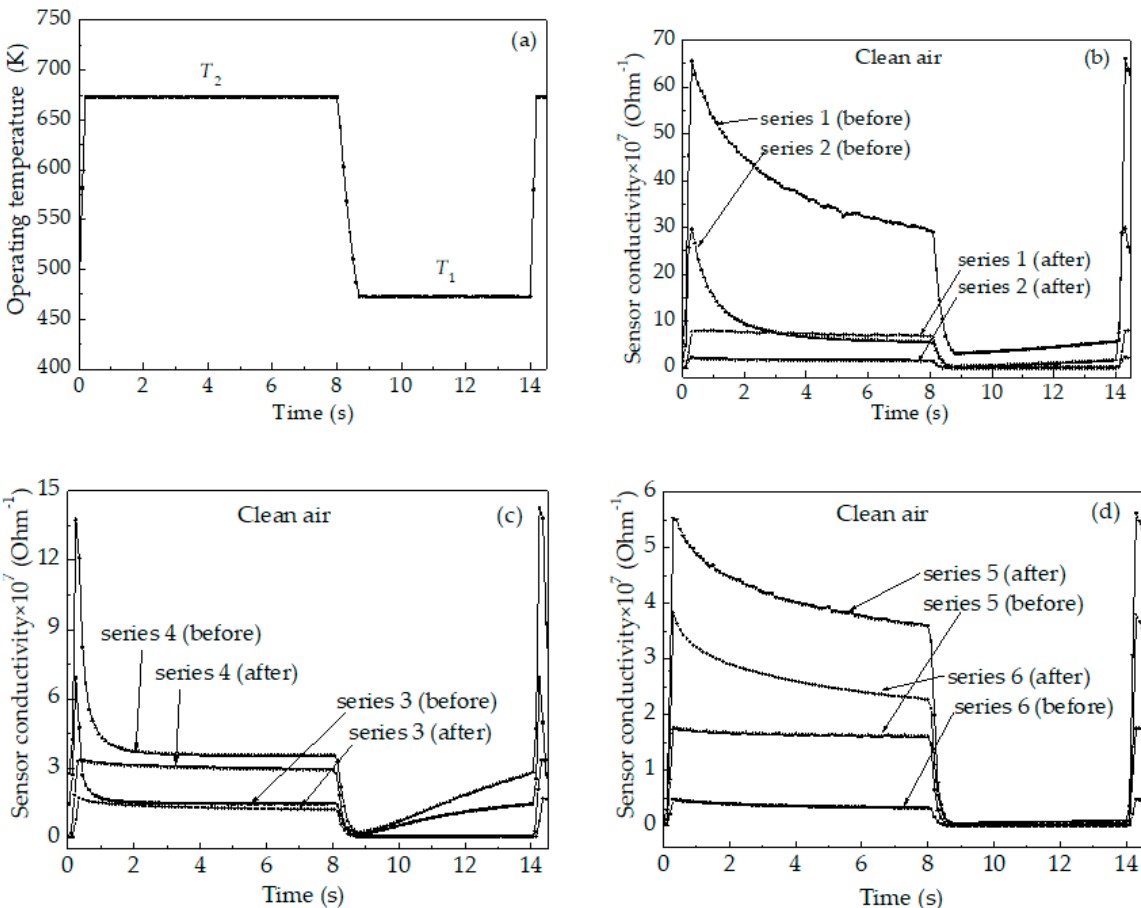

**Figure 12.** (a) The changing of operation temperature in thermo-cyclic mode and (**b–d**) the conductivity-time profiles in clean air in this mode before and after the tests of the sensors with the various additives.

The results of the evaluation of $e\varphi_s$ before and after the tests of the samples are provided in Tables 2 and 3. The dependence of the band bending on the operation time is shown in Figure 13. An analysis of these data shows a pronounced regularity: Before the tests for the samples of series 1–4, the values of $e\varphi_s$ varied from 0.41 to 0.52 eV in the sensors of series 5 and 6 with Ag + Y and Ag + Sc additives, the band bending reached 0.64–0.72 eV. During the tests, the most noticeable increase in the band bending is observed in the samples with the silver additives (series 2). In the sensors with the additives of Ag + Y (series 5), which were characterized by the increased stability of parameters, $e\varphi_s$ slightly increased.

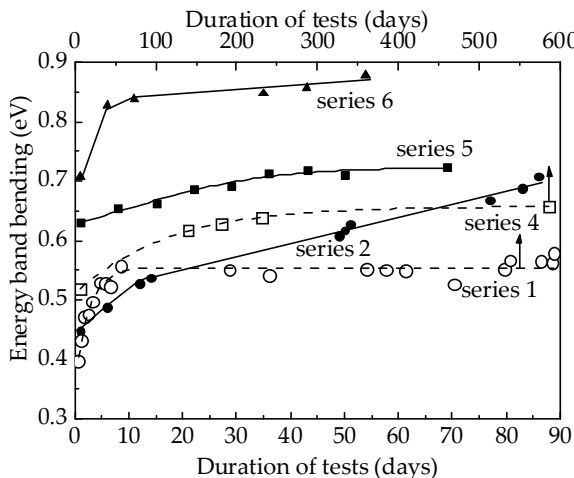

**Figure 13.** Dependencies of the energy band bending $e\varphi_s$ on the duration of the tests for the sensors of series 1, 2, 4, 5, 6.

### 4.3. Mechanisms of the Influence of Ag, Y, and Sc in the Volume of $SnO_2$ on the Properties and Stability of Hydrogen Sensors

It was assumed in [17,21] that the increase in the resistance of the films and the response to $H_2$ of the sensors based on the Pt/Pd/SnO$_2$:Sb and Au/SnO$_2$:Sb, Au films during prolonged exposure to hydrogen was due to the following effects. The atomic hydrogen released during dissociative adsorption of $H_2$ was able to interact with lattice oxygen. There was an increase in the density of superstoichiometric tin atoms, which are the centers of oxygen adsorption.

As shown by the data of the UV-vis spectroscopy and the layered AES introduced into the volume of films 1.2 at % silver segregated on the surface of nanocrystals in the form of metallic nanoparticles. The properties of the freshly prepared sensors of series 2 practically did not differ from the characteristics of the sensors of series 1 (Figure 10a and Table 2), but they changed most significantly during the long-term tests under periodic exposure to hydrogen. After operating for 60–70 days, the values of $R_0$ and $G_1/G_0$ increased by an order of magnitude (Table 3 and Figure 11).

It is important that unlike the samples of series 1, for which the responses to low (50–100 ppm) hydrogen concentrations practically did not change, the addition of silver led to a sharp increase in $G_1/G_0$ over the entire range of $n$ = 100–2000 ppm. To explain the role of Ag, the XPS data given in Figure 7c and Table 1 should be used, according to which the binding energy of Sn $3d$ and O $1s$ in SnO$_2$:Sb, Ag, Y and SnO$_2$:Sb, Ag, Sc films decreases compared to SnO$_2$:Sb, Sc. The authors [31] showed that the binding energy of Sn $3d$ and O $1s$ in SnO$_2$:Ag films decreases by 0.5–0.7 eV compared to pure SnO$_2$. As a result, Ag in the volume contributed to a more intensive process of interaction with lattice oxygen, when exposed to $H_2$, than in all of the other samples, and the density of the adsorption centers of chemisorbed oxygen increases. The increase in $N_i$ led to the most significant (almost 1.5-fold) increase in the band bending $e\varphi_s$ (Table 3 and Figure 13) during the testing.

The introduction of the yttrium (series 3) and scandium (series 4) additives contributed to some increase in the film resistance $R_0$. The XPS spectra of the studied films contain the core lines of rare-earth elements whose binding energies correspond to the metal $Y^0$, $Sc^0$ (Figure 7d,e and Table 1). In reference [25], hollow nanoscale tapes of tin dioxide with additives of 0.2 wt % to 0.7 wt % of yttrium obtained by electrospinning were studied as the acetone sensors. After annealing in the air for 2 h at 600 °C, mesoporous polycrystalline samples with the rutile structure were formed. Using XPS, it was established that the Y chemical state in SnO$_2$ did not correspond to the $Y_2O_3$ oxide, that is, yttrium oxide was absent and the $Y^{3+}$ ions were embedded in the SnO$_2$ lattice. In reference [26] for the as-prepared Y-doped SnO$_2$ (5 wt % and 10 wt %) nanostructure, both SnO$_2$ and $Y_2O_3$ crystalline phases are identified in the XRD diffraction patterns.

According to the reference data [46], the bond-breaking energy of the Y–O bond was $D_0^{298}$ = 718 kJ/mol and for the Sc–O bond it was $D_0^{298}$ = 689 kJ/mol. These values are higher than the bond breaking energy $D_0^{298}$ = 281 kJ/mol for the Sn–O bond. It can be assumed that during the thermal treatment of the thin films, yttrium and scandium segregated on the surface of the nanocrystals and formed stronger bonds with the lattice oxygen than Sn bonds with the lattice oxygen, hence the density of superstoichiometric tin atoms and oxygen adsorption centers increased. Therefore, during the long-term tests, the resistance and response to low hydrogen concentrations did not change. However, the action of yttrium or scandium was not sufficient to prevent an increase in the response at high concentrations ($n > 300$ ppm) of $H_2$.

In the films studied in this research, the joint introduction of 1.2 at % silver and <1 at % yttrium into the volume contributed to the formation of fine crystalline films and an increase in the $R_0$ and $G_1/G_0$ values in the freshly prepared sensors (Table 2). The response was stabilized over a wide range of $H_2$ concentrations during the long-term tests (Table 3, Figure 6). When the binding energy of the Sn–O bond was reduced in the presence of nanodispersed particles of metallic Ag, the $Y^{3+}$ cations penetrated (diffuse) from the surface into the volume and were distributed in the near-surface layers of the $SnO_2$ crystallites, thereby preventing the interaction of the hydrogen with lattice oxygen. The resulting "barrier layer" of the Sn–Y–O composition provided a reaction between the hydrogen and the chemisorbed oxygen only on the $SnO_2$ surface.

For the samples of series 6 with the Ag + Sc additives, the highest value of $R_0$ are characteristic (Table 2). Concomitantly, no stabilization of the parameters occurred over the entire range of hydrogen concentrations. Perhaps this feature was due to a lower value of $D_0^{298}$ as well as to a non-optimal ratio of the silver (1.2 at %) and scandium (3.8 at %) concentrations in the films.

## 5. Conclusions

Complex studies were performed of the nanostructure, composition, electrical, and gas-sensitive characteristics of $H_2$ sensors based on thin polycrystalline $SnO_2$ films produced by magnetron sputtering with dispersed Pt and Pd layers deposited on the surface and with the addition of Ag, Y, Sc, Ag + Y, and Ag + Sc in the volume. The additives of the silver and rare-earth elements in the volume of the films had a significant influence on the sizes of the tin dioxide nanocrystallites, the sensor resistance in pure air, the band bending $e\varphi_s$ at the grain boundaries, and the hydrogen responses over the concentration range of 50–2000 ppm. Particular attention was paid to the influence of the long-term tests on the properties of the sensors with the additives. Possible mechanisms of the effect of the additives on the properties of the sensors and the stability of the parameters during operation were considered.

In the Pt/Pd/$SnO_2$:Sb, Ag films with a concentration of approximately 1.2 at % of Ag, polydisperse metallic silver nanoparticles up to 10 nm and larger segregated on the surface of the tin dioxide nanocrystals of 30–60 nm in size and did not significantly affect the characteristics of the freshly prepared sensors. Concomitantly, in the process of testing, the increase in the resistance $R_0$, band bending $e\varphi_s$, and sensor response in the entire studied range of the hydrogen concentration (50–1000 ppm) was more significant than for all of the other sensors studied. Due to the decrease in the binding energy of Sn $3d$ and O $1s$, the Ag nanoparticles promoted active interaction of the chemisorbed hydrogen with the lattice oxygen and the formation of additional superstoichiometric tin atoms on the surface that were the centers of oxygen adsorption.

The Pt/Pd/$SnO_2$:Sb, Y (Sc) films contained large crystallites 200–350 nm in size in the Y-additives and 40–60 nm in size in the Sc additives. Since the binding energies of Y–O and Sc–O are higher than those of Sn–O, yttrium and scandium ions formed stronger bonds with the lattice oxygen during the thermal annealing of the sputtered films. As a result, the density of the adsorption centers of the chemisorbed oxygen increased, the resistance and the responses of the sensors increased, and partial stabilization of the parameters occurred: During the tests, $R_0$ and $G_1/G_0$ did not change at low concentrations of $H_2$ and the responses increased significantly only at $n > 300$ ppm.

For the freshly prepared Pt/Pd/SnO$_2$:Sb, Ag, and Sc samples, the highest value of $R_0$ and $e\varphi_s$, and higher values of $G_1/G_0$ occurred. However, there was no stabilization of the response in the entire range of hydrogen concentrations.

Of greatest interest are the characteristics of the sensors based on the Pt/Pd/SnO$_2$:Sb, Ag, Y films, into whose volume the additives of yttrium and silver were jointly introduced. In this case, the films are characterized by the increased resistances $R_0$ = 26–30 MΩ, ultra-high response values, and stabilization of $R_0$ and $G_1/G_0$ at 50–1000 ppm of H$_2$ during long-term tests. It can be assumed that there was a synergetic effect: In the presence of the silver nanoparticles, the Y$^{3+}$ ions actively formed strong bonds with the lattice oxygen, the hydrogen atoms interacted only with the chemisorbed O-ions, and the increase in the density of the oxygen adsorption centers and $e\varphi_s$ did not occur.

**Author Contributions:** Conceptualization, N.K.M. and A.V.A.; Methodology, E.Y.S., E.V.C., P.M.K. and S.N.N.; Software, E.Y.S.; Validation, N.K.M. and A.V.A.; Formal Analysis, N.K.M., A.V.A. and N.V.S.; Investigation, A.V.A., N.V.S., P.M.K. and S.N.N.; Resources, E.Y.S. and A.I.P.; Data Curation, N.K.M.; Writing—Original Draft Preparation, A.V.A. and N.V.S.; Writing—Review and Editing, N.K.M. and A.V.A.; Visualization, A.V.A. and N.V.S.; Supervision, N.K.M.; Project Administration, E.Y.S.; Funding Acquisition, E.Y.S. and A.I.P.

**Funding:** This work was performed as a part of the State Task of the Ministry of Education and Science of the Russian Federation (Task No. 3.9661.2017/8.9).

**Acknowledgments:** The authors are grateful to A. Biryukov and I. Shulepov for their assistance with the experiments and O. Vodyankina for helpful discussions.

**Conflicts of Interest:** The authors declare no conflict of interest. The funders had no role in the design of the study; in the collection, analyses, or interpretation of data; in the writing of the manuscript, or in the decision to publish the results.

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
