# Peer review of "Effect of Additives Ag and Rare‐Earth Elements Y and Sc on the Properties of Hydrogen Sensors Based on Thin SnO2 Films during Long‐Term Testing"

_coatings, doi:10.3390/coatings9070423_

Reviewer 1 Report

In this manuscript, the authors explain about “Hydrogen Sensors based on Thin SnO2 Films with Additives of Ag and Rare-Earth Elements Y and Sc”. Reported results are promising, however, it requires some refinement to improve the quality of the manuscript. I recommend a minor revision for this manuscript to publish in this journal. Please see the comments below

Main comments

 1.     Abstract: Please include important results like recovery-response time, selectivity and sensitivity towards target gas.

2.     How gas concentration has been calibrated? Please explain in detail.

3.     In the introduction, there is no discussion about scope/limitations of SnO2 and their heterostructures or with noble metal.

4.     About the gas sensor measurement, some parameters are missing like what is the SCCM of the gas? What is the applied voltage?

5.     It is better to include some strategies to improve selectivity and cite relevant references: Some suggested recent review article to cite in the intro. Part: Adv. Mater. 2016, 28, 795–831, Microchimica Acta, 185(2018) 213, Adv. Funct. Mater. 2017, 27, 1702168.

6.     It is very important to understand structural and morphology study for gas sensing application. Please include XRD and SEM image of pure and noble metal modified SnO2 samples.

7.     Please improve the quality of the figures, the visibility is blurred.

8.     What is the toxicity/threshold limit for Hydrogen? What is the limit for practical use? Authors should compare their sensing results with other materials in this range.

9.     Please include response plot of sensor curve for each concentration.

10.  Please provide response and recovery time Vs concentration with error bar.

11.  Please include the selectivity plot with sensor response curve with different oxidising and reducing gases.

Reviewer 2 Report

The paper presents the gas-sensing characteristics and parameters stability of H2 sensors based on thin nanocrystalline SnO2 films with dispersed Pt/Pd layers deposited on the surface and Ag, Y, Sc,  Ag+Y, and Ag+Sc additives in the volume. I have some major remarks that have to be addressed: 

Major:

Introduction:

lines 49 "nuclear power plant [], submarines [], 50 lactose intolerance [], the references are needed.

Materials and Methods:

line 75: the DC magnetron sputtering of SnO2 should be, at least, briefly presented, since references 11-16 are not OPEN ACCESS, and the readers have no idea about the proposed system. 

Line 76: commercial available Al2O3 substrates were need or home-made? more details are required here

line 76-77, is not clear: "The antimony..... [11]". What you want to say here?

Lines 78-79 a photo/ sketch is needed? 

Line 79: total pressure: do you mean base or working pressure?

Line 81: why 24 min. ?

Line 84: why 15 sec.? In the same process? did you change the targets? more details are needed!

Line 85-92 a photo of the sensor and sensor' heater would be very helpful

Line 122: a specially designed stand - it should be, at least, briefly presented here

Line 124: please define a recovery time as well

Line 129: a humidity sensor: name and model is needed

Reviewer 3 Report

The submitted manuscript presents studies on SnO2 thin films with different additives (Ag, Y, Sc) and their application as the hydrogen gas sensors. A lot of interesting and valuable experimental results are discussed in the paper. However, some points have to be clarified or improved. Therefore, this manuscript should be accepted for publication after major revision. The authors are kindly requested to answer following questions or consider remarks presented below:

1)      What is an exact mechanism of hydrogen gas interaction with SnO2 thin film? How does it change when different elements (Ag, Y, Sc) are introduced into SnO2 thin film? Did Authors investigated type of an electrical conductivity of pure as well as Ag, Y, Sc doped tin dioxide? When H2 comes into contact with a p-type semiconductor it dissociates and ionizes. The protons are adsorbed on the surface while electrons enter the lattice and recombine with positive holes decreasing number of majority carriers and reducing the current flow. When the protons are adsorbed on the surface of n-type semiconductor and the electrons enter the solid, the number of majority carries increases and the current flow enhances. Using of this effect for determination of conduction type seems to have a distinct advantage in the case of nanomaterials with high surface to volume ratio.

2)      What are detection limits of hydrogen for the prepared sensors? Authors provided only the minimum value of hydrogen concentration used in the experiments. There is a lot of detection limit definitions in the literature. One of them is given herein: The detection limit (Cmin), for the sensor, is the minimum amount of analyte that can produce a sufficiently different signal from the signal measured without a target gas. Typically, this is defined as the value of gas concentration for which a sensor response is greater than three times the standard deviation of the noise signal (σ).

Author Response

Round  2

Reviewer 2 Report

Authors have made all corrections and the paper can be accepted.

Reviewer 3 Report

Authors have made good revision of the initial submission. Therefore, the manuscript should be accepted for publication.